# The vicious circle of stereotypes: Teachers' awareness of and responses to students' gender-stereotypical behaviour

**Aleksandra Gajda**⬡* , **Agnieszka Bójko**⬡, **Ewa Stoecker**⬡

Department of Educational Sciences, The Maria Grzegorzewska University, Warsaw, Poland

⬡ These authors contributed equally to this work.
* agajda@aps.edu.pl

**Data Availability Statement:** The data from the transcription of the interviews as well as the data from the observational study were placed in the Mendeley Repository at https://data.mendeley.

## Abstract

The study was conducted using a mixed methods approach combining lesson observations and interviews with teachers. A total of 204 hours of observation in 34 classes of 7th and 8th graders (aged 13–14 in the Polish primary school system) were conducted to investigate teachers' behaviour that may exacerbate gender stereotypes and gender bias in the classroom. Moreover, the 25 female teachers conducting the observed lessons were interviewed to identify: (i) teachers' awareness of stereotypical behaviours of girls and boys during classes; (ii) teachers' awareness of possible causes of these behaviours; (iii) teachers' responses to these behaviours, including actions that could deepen gender stereotypes; and (iv) teachers' sensitivity to the gender polarised content of school textbooks. The results of the study show that teachers, although they are aware of the existence of gender stereotypes and declare their willingness to counteract them, tend to strengthen rather than eliminate these stereotypes with the strategies and actions undertaken. They have difficulty recognising possible reasons for the occurrence of stereotypical student behaviour and have little awareness of the gender-polarised content of school textbooks. The results of the study are discussed, inter alia, in light of the concept of the vicious circle of stereotypes and self-fulfilling prophecies in education.

## Introduction

As students in Organisation for Economic Co-operation and Development (OECD) countries spend an average of more than 7,500 hours in classes during their primary and lower secondary compulsory education [1], school, right after the family, is one of the most important socialisation environments in which girls and boys learn how to function in society in certain roles. As an important area of social life, school reflects all the mechanisms that one can encounter in society, including gender relations, social gender roles and even gender stereotypes [2]. These are complex sets of beliefs about individual characteristics, including competences, abilities, interests and roles performed by women and men [3]. It is believed that men and women differ in terms of achievement-oriented or social and service-oriented traits. The first set of traits is attributed to men, who are considered to be independent, decision-makers and

com/datasets/bn9vvjkj8h/1 [DOI: 10.17632/
bn9vvjkj8h.1]."

**Funding:** AG grant no. 2018/29/B/HS6/00036
National Science Centre https://www.ncn.gov.pl/
The funders had no role in study design, data
collection and analysis, decision to publish, or
preparation of the manuscript.

**Competing interests:** The authors have declared
that no competing interests exist.

aggressive, while the second set is stereotypically attributed to women, who are considered to be helpful, kind and concerned about others [4]. Masculinity is more often stereotypically associated with agency and instrumentality, while femininity is associated with communion and expressivity [5, 6]. Despite the passage of time, these social beliefs about gender roles have remained remarkably stable [7]. A characteristic feature of gender stereotypes is their dual descriptive and prescriptive nature. This means that they contain both elements of beliefs about the features that women and men have and beliefs about what features they should have [8]. Stereotypical descriptive beliefs make us perceive females and males as lacking the attributes necessary to succeed in areas occupied by the opposite gender. In turn, prescriptive beliefs discredit women and men who behave in a way that is inconsistent with how they should stereotypically behave [8].

Teachers, like other people, are not free from gender stereotypes [2]. Moreover, they are in positions of power due to their authority and influence [9]. Therefore, the functioning of girls and boys at school, conditioned by many factors, faces the attitudes of the teachers, who also shape the way young people function in terms of reflecting stereotypical gender differences [10]. Unfortunately, during teacher training in Poland, the topic of gender stereotypes is not addressed. There are bottom-up attempts to introduce interventions to counteract the limitations resulting from gender stereotypes; however, these interventions are usually proposed by non-governmental organisations (NGOs) as additional workshops or elements of informal education, and their number is marginal in relation to the actual need due to the scale of the phenomenon [11].

## School as an environment of socialisation to gender roles

The meaning of gender as opposed to sex began to be defined at the turn of the 1960s and 1970s as a relatively constant and unchanging construct consisting of cultural and social influences. Classic work by Goffman [12] indicates the behavioural aspects of being a man or a woman that constitute gender roles (or gender display, in Goffman's terms). Gender role is therefore a certain manifestation of normative behaviours, attitudes and actions appropriate for a given sex category [13]. In many respects, schools provide social experiences that reflect the socialisation experiences started by families and then acquired in relationships with the wider community and peer groups [14]. One dimension of this socialisation is socialisation with respect to gender roles. Taking the perspective of the socio-cultural basis of gender differences, we assume that, in the process of socialisation, students develop socially and culturally determined ways of reacting and behaving which to a large extent are based on their experiences in the school environment and in the process of social communication at school, where they spend significant time in their teenage years. One of the social mechanisms involved in the process of socialisation at school is the influence of significant persons, which largely comes down to imitation, modelling and identification with teachers [15].

There are several areas that shape the socialisation process at school: 'teacher-based dynamics, the formal curriculum, the school environment, peer dynamics, and teacher training and development as public policies that attempt to alter the role education plays in the emergence of gendered identities' [16, p. 4]. In our study, we look at the first aspect, teacher-based dynamics. As it is a complex process partly based on teachers' attitudes and beliefs, we used the mixed method in our study. 'Studies that focus on social dynamics call for both classroom observations and interviews with school agents regarding their daily practices' [16, p. 5].

Regardless of whether or not they are aware of their own beliefs and prejudices about gender roles, teachers are always their carriers, and due to their role, they pass these beliefs and prejudices on to children in the process of socialisation. Therefore, they can either participate

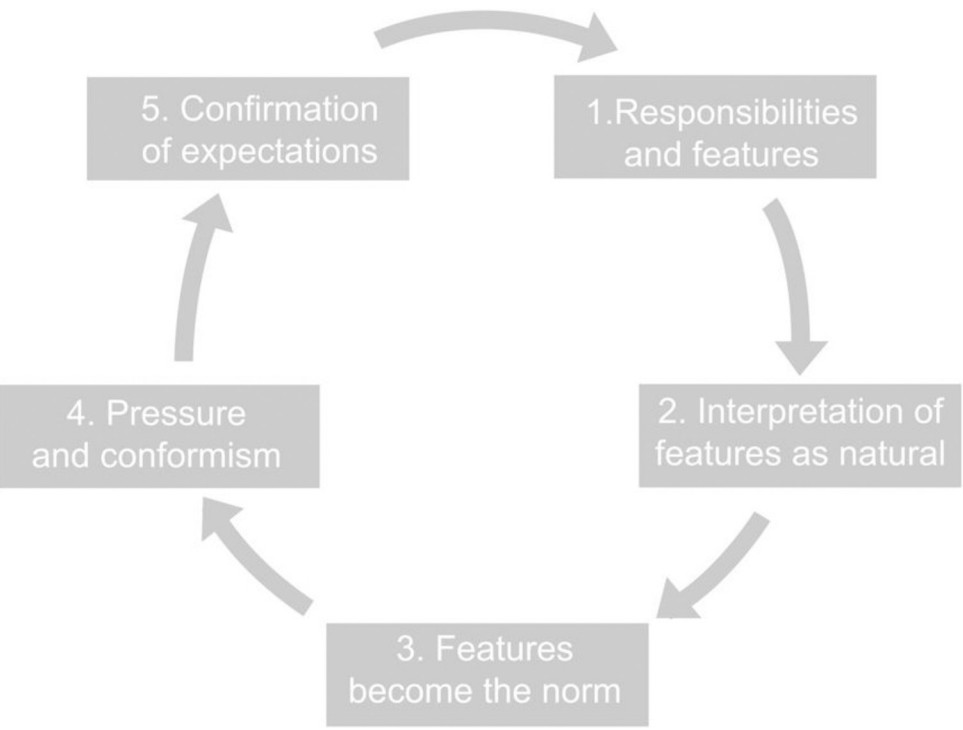

**Fig 1. The vicious circle of stereotypes.**

in the process of reproducing gender stereotypes or work to actively counteract it. An accurate visualisation of the process of reproducing gender stereotypes was proposed by Pankowska [17] in her theory of the vicious circle of stereotypes (Fig 1).

In the first step, in observing the social world, including the behaviour of people of different sexes, we can see that, for example, boys are more active and more involved in science lessons; they exhibit greater ease in acquiring science. In the second step, we recognise that the observed phenomena result 'from nature' and are not the result of numerous previous processes; thus, we interpret the 'linear mind' of boys as natural and assign it to all boys. In the context of gender stereotypes, which are based on binary opposites, girls are simultaneously said to lack a predisposition towards exact sciences (to fulfil the concepts of femininity and masculinity). In the next stage of the process, these features are normalised, and thus there is social conformism and pressure that in order to confirm one's masculinity and femininity, it is necessary to meet social expectations related to, for example, an interest in science. Since a failure to meet social expectations and destroying stereotypical images result in social sanctions, the vast majority of people conform to social ideals and fulfil social expectations, thus playing their gender role. Thus, the circle of the stereotype closes and the whole mechanism drives itself [15].

This self-reinforcing circle of stereotypes seems to correspond well with the definitions of a self-fulfilling prophecy present in social literature since the 1930s [17]. According to the theory above, if people identify situations as real, they are real in their consequences. It is therefore a false definition of a situation that triggers a new behaviour. In turn, this behaviour makes an initially untrue concept true [18]. A similar mechanism described in the psychological literature is called the interpersonal expectancy effect [19]. One person's expectations about another person's behaviour eventually lead to the behaviour occurring. Thus, with regard to

stereotypical behaviour of female and male students, a teacher expecting certain behaviour from girls and boys will more or less consciously differentiate their own behaviour towards students. As a result, female and male students will begin to manifest the behaviours expected by the teacher. This does not mean, however, that this spiral cannot be broken; it is possible, but it requires, firstly, awareness of the existence of this mechanism and, secondly, its disclosure—i.e. naming the consequences of stereotypical behaviour.

## Increasing gender stereotypes at school

Sometimes the different ways in which girls and boys function in the classroom seem to duplicate the stereotypes of gender roles existing in society [20]. Despite the fact that the school environment effectively recognises and adapts to differences in the functioning of girls and boys, steps are rarely taken to reduce them [21]. It has been observed that girls and boys have different attitudes towards work, aspirations and goals [22]. In the case of boys, the rejection of authority, an academic work ethos and high achievement have been observed [23], while girls have been said to be more mature and to use more effective learning strategies [24]. Girls are perceived as being more motivated, diligent, organised, hard-working and cooperative [22], while boys have a greater need to compete and lead [25]. It is also noticed that girls are characterised by lower self-confidence and lower self-esteem during lessons [26, 27]. They need reassurance of the value of their own work more often and they are more likely to be assisted by a teacher [28], while boys are usually more independent and convinced of their abilities [25], especially in the exact sciences. This is interesting because boy culture is less oriented towards learning than girl culture. It turns out that boys' school achievements increase as the number of girls in school increases [29, 30].

Yet another source from which students derive information about gender stereotypes is the content of school textbooks. As shown in worldwide research, reconstruction of the representation of women and men in textbooks contributes to the deepening and consolidation of stereotypical behaviours and gender roles among students. The results, regardless of where studies were conducted, are very similar. A high level of gender stereotyping is often reported [31, 32], as well as an overwhelming number of male characters [33]. Usually, it is men who have higher-paid jobs and better employment [34], while women and girls usually do household chores [35]. Boys are presented as brave, bossy and enterprising, girls as submissive, gentle and insecure [36]. The results of Polish research confirm this trend, pointing to the presence of a greater number of male characters, earning more and taking up professions related to power, while the place of women is presented mainly in the kitchen or at home with childcare [37]. Male characters usually deal with exact sciences, study and practise professions that require mental work, while female characters choose activities related to artistic activities. Unlike men, they are not decision-making or enterprising, and more often require assistance with various activities [38].

Also, as briefly suggested earlier, with teachers themselves often unknowingly deepening gender polarisation [39], a self-fulfilling prophecy may arise [28] in which the behaviour of girls and boys is reinforced by teachers' messages and attitudes, resulting in the duplication of well-established patterns of student behaviour, limiting the possibility of changing the way of thinking and functioning. Both lesson observations and student interviews have confirmed that there is a different distribution of interactions between the teacher and students depending on their gender. In response to a less learning-oriented boys' culture, teachers sometimes unconsciously reinforce male dominance in the classroom by focusing their attention on boys, explaining instructions directly to them in detail, and answering their questions more often [27, 28]. Girls are often overlooked when interacting with their teacher, and their actions are

often unnoticed or undervalued [40, 41]. Teachers seem to ignore disruptive boys' behaviour or perceive it as typical for them [42] while significantly more is expected of girls in terms of both behaviour and performance [43]. Moreover, teachers more often look for the reasons for boys' successes and failures in their abilities, and for girls' in their effort [25].

These differing teacher actions and expectations have negative consequences for both girls and boys, especially in terms of the subject being taught. Girls' low performance in science is explained by lower teacher expectations [44]. On the other hand, for boys, there are lower expectations in terms of their achievement in humanities and languages [45], which coincides with their lower achievement in this field. In many cases, there is even justification for and tolerance of boys' low achievement in language subjects, while girls' high achievements in humanities are often underestimated and trivialised in line with the belief that working in these subjects comes naturally to them [41, 46]. This leads to a stereotype threat phenomenon consisting of a worse performance of tasks related to the activation of a negative stereotype about the low ability level of an individual's own group [47]. Polish research in which primary and middle school students took part indicated that the stereotype threat has consequences in the form of a worse result in knowledge tests, lower school grades, and expectations of failure in the future and in the long term beyond the scope of particular scientific field [48, 49]. Moreover, it turns out that teachers themselves can strengthen the described phenomenon [50]. For example, girls who are taught by teachers with traditional views on gender roles score lower on standardised mathematical achievement tests, while boys are not affected [51].

Teachers are more and more often aware of the presence of gender stereotypes in the classroom, and even despite the lack of appropriate training, they make attempts to eliminate them [52]. It turns out, however, that they are not entirely free from their own stereotypical beliefs, which usually concern higher mathematical skills among boys [10] and higher language skills in girls [53]. The relative lack of awareness among teachers of the profound nature of gender stereotypes and their impact on the functioning of students is also confirmed [54]. Although teachers in interviews declare the need to reduce gender polarisation in textbooks [41], direct observation of classes shows that they actually ignore gender-biased representations occurring in the teaching content [55]. The reported results are not unambiguous, but it seems that even a certain level of awareness of gender stereotypes does not protect against the unconscious deepening of their impact. Considering the above, four main goals of this study were established: to examine (i) teachers' awareness of gender stereotyped behaviours of girls and boys; (ii) teachers' awareness of the possible causes of these behaviours; (iii) teachers' actions in response to these behaviours, including those that may deepen gender bias and gender stereotypes during classes; and (iv) teachers' awareness of the existence of gender-polarised content of school textbooks. Therefore, the study seeks to examine the challenges faced by teachers in the context of accurately identifying and responding to gender stereotypes that emerge in the teaching-learning process.

## The present study

Literature on the differences in the behaviour and functioning of girls and boys at school is quite extensive, covering their school performance [56–59], preferred school subjects [60–62], gender-stereotypical classroom behaviour [51, 52, 54], and differences in the way teachers treat female and male students [25, 28, 43, 63]. However, there are few analyses linking teachers' beliefs and declarations regarding gender issues with observations of their actual activities during classes. This issue requires a closer look. Therefore, to fill the data gap in this area, our study examines the perception of gender issues of teachers working with 7th and 8th graders

(aged 13–14 in the Polish primary school system). The following research questions were formulated in line with the objectives of the theoretical framework:

1. Do teachers treat girls and boys differently in terms of providing assistance and encouraging independence, encouraging cooperation and competition, and praising and criticising their behaviour and skills?

2. Do teachers notice (gender-stereotyped) differences in behaviour between girls and boys?

3. What causes of gender-stereotyped behaviour do teachers identify?

4. What is teachers' response to gender-stereotyped behaviour in students?

5. Do teachers notice gender-polarised content in school textbooks?

To address accuracy and reliability [64] in our study, we used a mixed methods approach combining the results of teachers' interviews with the results of lesson observations, which were conducted to explore teachers' behaviour that reduced or deepened students' gender-stereotypical behaviour. As indicated earlier, research on social-based dynamics (here, teacher-based) gains in reliability and validity when using a combination of the above research methods [16] which enables comparison of teachers' observed behaviour towards students with their subjective experience manifested during the interview.

## Method

### Study procedure

The study consisted of two independent stages: lesson observation and interviews with teachers. The first, observational stage aimed to identify teachers' behaviour that may exacerbate gender stereotypes and gender bias in the classroom (research question1) and included 34 classes in 17 randomly selected schools, located in small, medium and large towns in central Poland. Administrative units were selected at random; then schools were drawn from the above units and, in the last step, one class from year 7 and one class from year 8 were randomly selected in each school. In order for the class to take part in the observational study, written consent to participate in the study had to be given by all students' parents or legal guardians. In the absence of consent from everyone, the next class was drawn. In order to maintain the anonymity of the study participants, no personal data on the students participating in the classes were collected. Six lessons were observed in each class (three mathematics lessons and three Polish language lessons), which gave a total of 204 observed lessons. Such a number of hours of observation for each class was aimed at familiarising students and teachers with the researchers and limiting the impact of the presence of the researcher on the functioning of the teacher and students [65]. Moreover, in order to ensure the objectivity of the study, each lesson was observed by two researchers trained by the first author of the article with regard to the study's subject. Each lesson was audio recorded, and during the lesson the researcher filled in an observation sheet, marking the emerging behaviour of the teacher relevant to the study. During the lesson observation, the teachers' behaviour in relation to students was coded, corresponding to eight categories: (i) encouraging cooperation or (ii) competition; (iii) appreciation of skills or (iv) behaviour; (v) criticising skills or (vi) behaviour; and (vii) encouraging independence or (viii) providing assistance. Behaviours classified into individual categories were added up, keeping the division between teachers' messages addressed to boys and girls separate. To reduce the impact of awareness of the purpose of the study on the functioning of teachers, they were informed that the purpose of the observation was to identify different functioning of girls and boys during the classes in contact with the teacher. After the observation

was completed, the observation sheets were compared; any disputes were settled on an ongoing basis. The first author of the article, after the completion of the observational study, reviewed all the recordings of the lessons, comparing them with the observation sheets.

The second stage of the study aimed at answering the four main research questions highlighted above. In this stage, the teachers of mathematics and Polish language conducting the observed lessons were invited to take part in individual interviews. In order to examine teachers' awareness of gender-stereotyped functioning of students and to trace the strategies they undertook in relation to these behaviours, semi-structured interviews were carried out. The interviews were conducted online and were completed over a three-month period. Each conversation, with the verbal consent of the interviewee, was audio recorded and lasted an average of one hour. In order to ensure anonymity in the study, teachers' sensitive data were not used, and the data obtained from the interviews were identified using appropriate codes.

The interviewees were asked to answer the questions as broadly as possible. The questions included in the interview questionnaire were formulated on the basis of research on gender stereotypes in a school environment [66–68], teachers' attitudes towards students [43, 63, 69] and gender bias analyses in school textbooks [37, 38]. The final version of the interview questionnaire included ten questions corresponding to the four main research questions presented in Table 1. For the sake of good data quality, honesty and authenticity, as suggested by Lincoln and Guba [70], we conducted interviews based on several key elements. The interviews were conducted by two researchers, being the first and third authors of this article. While one of the researchers conducted the interview according to the prepared questionnaire, the other kept notes and took care of the good quality of the audio recording. The recordings were then transcribed verbatim and, together with the notes, made available to the interviewees for checking.

## Participants

In both stages of the study, 25 female teachers (12 mathematics teachers and 13 Polish language teachers) participated. The average age of the surveyed teachers was 42.6 years, their ages ranging between 31 and 47 years. The average work experience among the interviewees

**Table 1. The main research themes and the corresponding questions in the interview.**

| The four main research questions | Questions included in the interview |
| --- | --- |
| 1. Do teachers notice (gender-stereotyped) differences in behaviour between girls and boys? | *How do boys and girls work during your lessons? Do you notice a difference in their work and behaviour depending on their gender?* |
| | *What are the results of girls and boys in the subject? Do these results differ according to students' gender?* |
| | *Do you notice any typically girlish or typically boyish behaviours in students?* |
| | *Do students prefer the subject depending on their gender?* |
| 2. What causes of gender-stereotyped behaviour do teachers identify? | *Can you think about possible causes of the differences between the boys and girls you described in the previous statements?* |
| 3. What is the teacher's response to gender-stereotyped behaviour in students? | *What is your reaction to such different behaviours of girls and boys?* |
| | *Do you value these behaviours in some way? Do you find them problematic/neutral for the teaching-learning process?* |
| 4. Do teachers notice gender-polarised content in school textbooks? | *Do you notice how male and female characters are portrayed in the textbooks? Are there any differences between them in your opinion?* |
| | *Can you recall what characteristics, interests, activities, etc. are attributed to male and female characters in textbooks?* |

was 16.8 years. The shortest work experience was six years, and the longest was 21 years. It was not assumed that only women would participate in the study. However, it turned out that there was no male teacher of mathematics or Polish language in the observed classes. Some teachers taught in more than one class—hence the difference in the number of observed classes and the number of teachers surveyed.

## Ethical statement

The study was approved by the ethics committee of the Maria Grzegorzewska University (approval number 43/2021). The study was prepared in accordance with the ethical standards applicable to the social sciences [71]. Teachers gave oral consent to participate in the study, which, in line with the ethical principles of social research, is equivalent to written consent for adults, while parents provided written consent for their children's participation in the observational part. Following the above ethical standards, during both the first and second stages of the study, participants were informed of their ability to withdraw from the study at any time without giving any reason. In order to maintain the anonymity of participants, confidential information about participants was not collected at any stage of the study. After completion of the second stage of the study, participants were provided with transcripts of the interviews for viewing.

## Data analysis

Observations of teachers' behaviour during classes were analysed using a quantitative methodology, using the SPSS IMAGO 27 package. The unit of analysis was a single recorded behaviour of the teacher towards a male or female student; these were then classified into the appropriate category (see Table 2). To maintain the reliability between the two encoders, after observing and coding the first six lessons, the intercoder reliability was calculated using Cohen's kappa (Table 2). After conforming, the coders encoded the rest of the data. The total number of observed behaviours was N = 4,127.

The interview analysis was based on grounded theory. The purpose of this research approach is to collect and analyse data based on the indicative-interpretative paradigm. It aims to understand theories and patterns of human behaviour in the social context, abandoning existing theories [72, 73]. The study used elements of grounded theory to examine teachers' awareness of students' gender stereotypical behaviour and to track their strategies in response to observed behaviours. As the basis of grounded theory is the assumptions that people actively form of the world in which they exist through the process of symbolic interactionism, the use of this research approach in this study seems justified [72]. The constant comparison method was used during data analysis: the themes appearing in teacher interview data were compared until the data were saturated and no new themes were recorded [73, 74]. Consistent parts of statements representing one thread were assumed as code units. The analyses were performed using MAXQDA qualitative data analysis software. Two coders, the first and second authors of the article, dealt with data encoding. In the first stage, analysis of all interviews was carried out and the fragments important for the research objectives were given codes from which the initial list of themes was created. At this stage, 68 codes within 13 categories were identified. All interviews were then re-analysed, comparing the descriptions assigned to the preliminary codes, and reducing and combining themes that were related to the same content. The final version of the code tree included four main themes, 11 subcategories and 53 codes (Table 3). In the last step, the third independent coder (the third author) re-analysed all the created codes and data to check the interpretation of the interviews. The third coder re-read all the interviews with the marked analysis units and their assigned codes, and then assessed whether

**Table 2. Code categories of teachers' behaviour and interrater reliability.**

| Category | Description | Example | Cohen's kappa |
|---|---|---|---|
| Encouraging cooperation | All forms of encouraging children to cooperate with each other, regardless of gender. | *Creating situations in which children can cooperate and exchange skills* | 0.84 |
| | | *Students learn from each other* | |
| | | *Students create a product together* | |
| Encouraging competition | All forms of encouraging children to compete with each other, between genders, and irrespective of gender. | *Dividing into opposite-sex or same-sex groups and creating situations of competition for results between groups* | 0.88 |
| Appreciation of skills | All forms of praise regarding students' skills, their work in the classroom, perseverance, their abilities, products, etc. (in terms of skills and work performed). | *The teacher praises the good result obtained in the test, emphasises the high abilities of the student* | 1.0 |
| | | *The teacher praises the student for his/her extensive knowledge and correct answer* | |
| Appreciation of behaviour | All forms of praise for the behaviour of students, regarding, for example, being polite, quiet, diligent, attentive, etc.; verbal or written praise (only in the context of behaviour, not work results). | *Praise for being a diligent student and keeping the notebook in order* | 0.95 |
| | | *Praise for being polite and well mannered* | |
| Criticism of skills | All forms of criticism regarding students' skills, their work in the classroom, their abilities, products, etc. | *Criticism of the student's ability to solve the task at the blackboard* | 0.83 |
| | | *Criticism of the student's knowledge while taking the test* | |
| Criticism of behaviour | All forms of criticism of students' behaviour: comments on behaviour, conversations and inappropriate comments from students; verbal or written criticism (written reprimands given to students). | *Written reprimand given to the student for disrupting the class* | 0.91 |
| | | *Verbal rebuke to students for talking in class and lack of discipline* | |
| Encouraging independence | All manifestations of encouraging independent work: motivating students to actively approach problems, strengthening students' independence, etc. | *The teacher encourages the student to perform a difficult task, emphasises his or her ability to do it* | 1.0 |
| Providing assistance | All forms of the teacher providing assistance: doing tasks, rewording the theory and instructions for completing the task, relieving students in answering questions, etc. | *The teacher performs the task for the student* | 0.89 |
| | | *The teacher answers the question instead of the student* | |

the assigned codes were relevant to the content of the analysis units. Each fragment was rated on a seven-point scale (1 = no agreement, 7 = high agreement) [75]. The average assessment of compliance of the codes with the content of the units of analysis was 6.5; therefore it was assumed that there was no overinterpretation of the data in terms of the representation of repeated patterns of experiences reported by teachers.

# Results

## Observational findings

The results of the observation analyses were presented in a quantitative form by conducting a series of T tests of the significance of differences between the number of observed behaviours of the teacher in each of the six examined categories.

Fig 2 shows the percentage distribution of the teachers' message categories to male and female students. According to the data presented in Table 4, the greatest differences were observed for two types of messages: over 95 per cent of encouraging cooperation messages were directed towards female students ($t(62) = -3.222$; $p = 0.003$), while over 80 per cent of encouraging competition messages were directed towards male students ($t(46.648) = 2.239$; $p = 0.03$). These differences turned out to be statistically significant: teachers significantly more often strengthened the ability to cooperate in girls, and in the case of boys, the will to compete. Boys were also criticised for their behaviour significantly more often than girls (64.3% for boys vs 35.7% for girls; $t(52.979) = 1.993$; $p = 0.05$). It is also worth emphasising that

**Table 3. The final number of themes, categories and codes.**

| I. GIRLS / BOYS DIFFERENCES | II. POSSIBLE CAUSES OF DIFFERENCES |
|---|---|
| **Lack of awareness of the differences** | **The reason is hard to find** |
| **The need to be perfect (girls)** | **Social factors** |
| Modesty, belittling your own abilities | Awareness of your own stereotypical thinking |
| Following a known pattern of action | The influence of electronic media |
| Diligence and hard work | Early school education |
| Need for consultation, uncertainty about self-efficacy | Upbringing at home |
| Being active and being accurate | Stereotypes rooted in culture |
| **Emotionality and relationships (girls)** | **Biological / psychological factors** |
| Tired of following the rules | The brain works differently |
| Emotional problems, complexity | Maturity |
| Artistic sense | Girls have better fine motor skills |
| Psychological violence | Differences in temperament |
| The need to be part of a group | **III. RESPONSE TO THE OBSERVED DIFFERENCES** |
| **Humanities / science** | Students and equal rights |
| Polish language as a domain of girls | Eliminating stereotypes |
| Polish language—outstanding boys as an exception | Encouraging students to go beyond the patterns of action |
| Mathematics as a domain of boys | Conversations about gender stereotypes |
| Mathematics–outstanding girls as an exception | Encouraging agency |
| **The need to demonstrate the result (boys)** | Pointing out examples, role models |
| Less refined and specific works | **Strengthening stereotypes** |
| Action outside the box, creativity | Favouritism |
| Achieving quick results | Beautiful girls and bullying boys |
| **Rationality and leadership (boys)** | Own division into gender teams during classes |
| The need for agency, efficiency and independence | Girls are careful and delicate |
| Educational problems and rebellion | Men repair and DIY |
| Competition and physical violence | The women do housework |
| Need to lead, leader | Keeping boys from acting |
| Boys' solidarity | **IV. TEXTBOOKS** |
| Courage and domination | Pro-equality attitude |
| **Carelessness (boys)** | Awareness of non-stereotypical content |
| Outstanding means disliked | Awareness of gender polarisation |
| Carelessness, lack of accuracy | Lack of attention to stereotypical content |
| Lack of learning discipline | Gender polarisation is downplayed |

despite the lack of statistical significance, teachers interacted more often with boys than with girls in each of the analysed categories of behaviour (with the exception of encouraging cooperation, as mentioned above). More than half of the messages were addressed to male students.

## Interview findings

The analysis of the data obtained from the interviews was based on the four main themes mentioned above. The order of the themes corresponds to the order in which the threads were discussed during interviews with teachers. The most extensive topic that aroused the greatest interest among the interviewees was the differences in the behaviour and functioning of boys

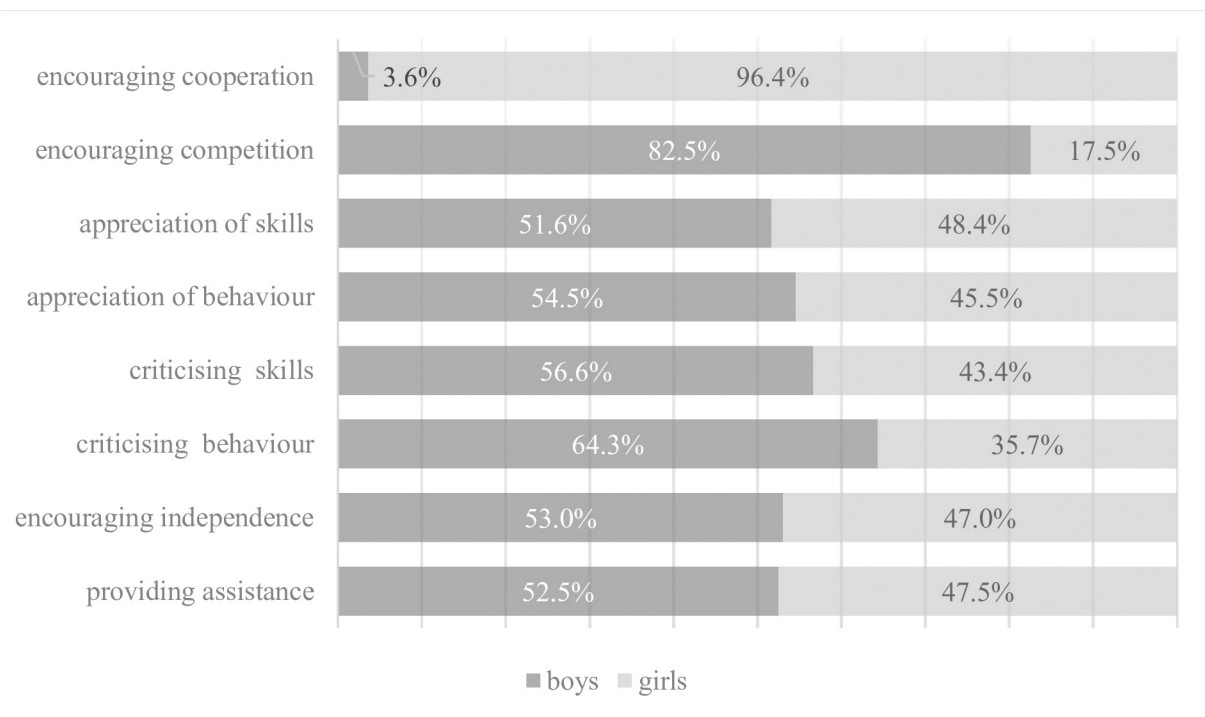

**Fig 2. Distribution of the teachers' message categories to male and female students.**

and girls during classes, which can be described as stereotypical in terms of gender. This area is discussed first.

**I. Awareness of the existence of stereotypical traits and behaviours presented by male and female students.** The question of whether teachers observe differences in the behaviour and functioning of boys and girls during lessons has been discussed extensively. Generally, teachers were aware of a number of differences that could be defined as stereotypical behaviour among male and female students. However, there was also a group among the surveyed teachers who declared that they did not observe such behaviour among the students. They declared that boys and girls worked similarly during the lessons, achieving neither better nor worse results in any of the subjects analysed. Any differences that did occur were most often explained as a matter of chance or personal predisposition. This group of teachers also

**Table 4. T test results and number of behaviours observed in each category.**

| Observed category | N boy/girl [a] | t | df | p | SE | 95% CI |
|---|---|---|---|---|---|---|
| Encouraging cooperation | 5/134 | -3.222 | 31.492 | .003 | 1.251 | [-6.58; -1.48] |
| Encouraging competition | 52/11 | 2.239 | 46.648 | .030 | .572 | [.13; 2.43] |
| Appreciation of skills | 569/533 | .233 | 62 | .817 | 4.830 | [-8.53; 10.78] |
| Appreciation of behaviour | 42/35 | .579 | 62 | .565 | .378 | [-.54; .97] |
| Criticising skills | 64/49 | .628 | 62 | .532 | .746 | [-1.02; 1.96] |
| Criticising behaviour | 319/177 | 1.993 | 52.979 | .051 | 2.226 | [-.028; 8.90] |
| Encouraging independence | 651/577 | .561 | 62 | .577 | 4.121 | [-5.93; 10.55] |
| Providing assistance | 477/432 | .426 | 62 | .671 | 3.300 | [-5.19; 8.00] |

[a] The first column contains the number of observed teacher behaviours in relation to girls and boys separately.

declared that educational problems possibly occur in both boys and in girls and that it is in no way dependent on the gender of the student. Teachers argued that the gender gap is slowly blurring. Interestingly, they only gave examples of girls resembling boys and no examples of boys resembling girls.

> A lot of girls who are... tomboys, we laugh often, that they can play football with boys (...) (T25)

*Stereotypically feminine and masculine school subjects.* The context of the division of fields of study into those "typical" for girls and boys appeared in almost every teacher's statement. What the teachers' declarations had in common was that girls were assigned preferences for and better skills in the humanities and boys in mathematics.

Identifying gender differences in this respect, teachers quickly passed over them, stating that 'Girls, maybe more humanists, like to tell stories more. Women scientists are rare' (T1). Girls were associated with people who have greater ease in expressing their feelings, which facilitates their studies in the humanities and gives them greater freedom: 'However, girls are more attracted to literature, and boys, however, to science, mathematics and computer science' (T6). The humanities were, in fact, indisputably associated with girls, who were considered to have higher reading skills and a greater ability to express themselves. Teachers noticed their greater level of participation in the lessons and remembered their high skill level, while boys were considered not to be interested in humanities. In the eyes of teachers, girls perceive the surrounding reality differently and see more, unlike boys, who strive for concise statements, without the need to talk about their own feelings. Boys distinguished by their humanistic achievements or who are interested in activities related to this field were considered exceptional.

In turn, mathematics was almost unanimously recognised as the domain of boys; however, in this case, teachers more often recognised that boys and girls work differently. At the same time, they declared that girls and boys ultimately achieve similar results. There was even the perception that after the end of primary school, 'girls are afraid of these more technical areas. They are more artistic, they choose law or journalism, and boys prefer strictly technical subjects, computer science, automation, robotics and so on' (T13).

*Emotionality and relationships.* Another area where teachers noticed gender differences was the emotional sphere of girls as opposed to boys' rationalism. Girls were generally considered to be more 'complicated'. This is a challenge for teachers, and it makes it difficult for girls to benefit fully from the educational process: 'She can solve the task at a very good level when she is alone, and when she is in the classroom, she practically falls to a quite low level' (T1). On the one hand, girls were capable of creating good, advanced ideas; on the other hand, they were still treated as immature by their parents. They experienced difficulties related to the regulation of emotions and the search for their own identity, with a strong need for a relationship with the teacher:

> (...) in adolescence she rebelled so much, she began to look completely different, she manifested her personality with her appearance, and the teacher who took her over in junior high said that it was impossible to force her to do anything (T7).

A characteristic feature of female students was the strong need to be accepted by the group and by the opposite sex, which, as teachers emphasised, was accompanied by high, unregulated emotions. Sometimes girls were excluded from their peer group because of their excessive maturity or, on the contrary, their insufficiently developed social skills. Girls' disputes often

also concerned social relations, 'expectations that I want to have an exclusive friend' (T8), were highly emotional, and they 'tease[d] each other more' (T19).

*Uncertainty and striving for perfection.* Girlish behaviour was dominated by modesty and the need to be perfect. In the teachers' opinion, the girls tended to undermine their own achievements and skills: for example, 'when I praised her painting and said it was very good, she had such a feeling that "er, not at all" as if she thought that I was saying it only to it be nice' (T8). When they achieved success, they seemed to be embarrassed and intimidated by this fact; they tried to explain it by the lack of competition or the low level of difficulty of the task. Girls were often perceived as insecure and withdrawn.

They had a strong need to confirm their abilities—'usually (girls) ask for verification of their behaviour' (T11); they often expected and asked for feedback, not being sure of the quality of their own ideas. On the other hand, they did not hesitate to ask for help, thanks to which they were able to verify possible errors in thinking faster. The uncertainty of their own abilities made the girls choose standard paths to solve tasks or problems—'they check everything very meticulously, step by step' (T3), usually following the instructions carefully, fearing to make a mistake. This was especially visible during maths lessons. An undoubted advantage of such a manner of working, referred to as 'such hard work' (T19), was, according to the teachers, the lower number of errors in the results. The diligence shown with respect to their work—also visible, for example, when taking notes,—made it easier to find information faster when necessary: 'Girls always have everything nice, aesthetically arranged, everything as it should be' (T1). The girls' strategy for action, based on regularity, diligence and following a well-known pattern, worked well enough to achieve satisfactory school achievements and pass subsequent exams. Unfortunately, as the teachers pointed out, there were also disadvantages to the above-mentioned way of functioning. One of them was the disproportionate effort put into solving the tasks, sometimes resulting in 'getting lost' in the calculations:

> The girls just get lost in it; they count and they count, and they don't see the goal they want to reach. And boys present it graphically somewhat faster; girls are less able to cope with it (T15).

The teachers also pointed to another disadvantage of the tendency to follow a pattern, which was avoiding combining. This, in turn, led to more frequent abandonment of answers by girls who were not sure: 'girls give up faster. They say: "I couldn't do it", and they skip this task' (T23).

*Rationality and the need for agency.* The functioning of boys, on the other hand, was dominated by rational elements, manifested in various areas. They were straightforward in action, which also translated into the ability to make decisions efficiently, and they were characterised by pragmatism of action, which was especially visible when carrying out specific commands. They preferred activities that ensured the simplest path, 'as short as possible, by a short path and to the goal' (T1), preferably devoid of emotional charge. The teachers emphasised a strong need for agency in boys, which was manifested, inter alia, by their ability to act outside the usual pattern. Male students were more likely to avoid writing down series of mathematical calculations, preferring shortcuts and counting in their mind. It was important for them to get the correct result in the shortest possible way: 'They would tell me: why write it down? You can calculate it in your head' (T3). Although this feature hindered the didactic process—'this is their disadvantage, but in turn they have better ideas' (T3)—it also led to the development of boys' ability to create unusual, creative solutions to problems. Teachers often emphasised male students' non-stereotypical thinking skills—'boys always have a thousand ideas' (T7); 'they are up to something more often and it helps them more often' (T23)—especially in maths lessons.

On the other hand, in Polish language lessons, the boys' shortcuts did not work so well. Teachers emphasised the reluctance of male students to write essays in which the imagination or reading knowledge should be used: 'there are no unnecessary sentences or words there' (T25). According to teachers, boys more often declared their reluctance to learn the Polish language, arguing that with the lack of ideas for written works, they saw no sense in it, 'because for them it (brief statement) is good and everything is needed quickly, briefly, concisely and to the point' (T14). The desire to achieve a result quickly made the written forms they created short, specific and sometimes even sterile, which translated into lower results. Boys were much more frequently characterised by a lack of systematic and disciplined learning. In teachers' opinions, they often had great intellectual potential but used it very little, neglecting learning if it required greater effort from them. Therefore, more often than girls, they took advantage of compensatory classes, and they were also more likely to suffer from dyslexic disorders. According to teachers, their careless attitude was present not only during the lessons when carrying out tasks but also at the stage of preparing for the lesson or supplying the necessary materials.

It follows from the above opinion that the boys were rarely "good pupils" who were always prepared for lessons. And if they were, they were usually rejected by their peer group or were disliked. Too much participation during classes, high achievement or interest in learning did not go hand in hand with acceptance by a group of male peers. However, attitudes proving boyish resistance to the applicable rules and a willingness to skip school requirements were far more frequent: 'boys want less, prefer someone to get things done, someone replies, and they somehow survive on the girls' backs until the break' (T5). The necessary tasks, if any, were performed at the last possible date and followed the path of least resistance, in teachers' judgement.

*Leadership and domination*. According to teachers, boyish rationality also manifested in the acceptance of the social order, which functioned very efficiently; they had a strong need for leadership. In a group of boys, there is usually a leader who is accepted by peers, and usually his ideas are implemented. Teachers emphasised that the disputes arising among boys, their high ambitions and tendencies to compete with each other did not affect the sense of solidarity and the ability to function in a group.

One of the symptoms of dominance among boys is the more frequent tendency to resolve conflicts through physical violence. The teachers took it for granted, giving it a kind of silent consent: 'when it comes to bullying, yeah, guys know they fight' (T19). The teachers also emphasised that 'there is a gender division in this area because there is no physical violence (. . .) when it comes to girls' (T19). Characteristic in teachers' opinions was the rebellious behaviour of male students, which was observed much more often than similar behaviour among female students: 'boys at some point rebel more than girls' (T8). The rebellious behaviour most often described concerned a reluctance to comply with the existing educational rules or questioning the legitimacy of performing tasks related to the topic of the lesson: for example, 'he just comes and demonstrates I will not write, I will not do, I will not read, everything is stupid and so on' (T25). According to the teachers, non-conformism, typical of boys, was troublesome because on the one hand it made it difficult for the teacher to conduct lessons and for students to learn, and on the other hand, teachers incurred high emotional costs when trying to respond to it. Interestingly, teachers repeated many times that the boys described as the worst rascals were ultimately liked by them: 'that rascal of mine came to hug me. Just when they handed me flowers, it was so nice' (T8).

**II. Identification of the causes of gender-stereotyped behaviour.** A small number of teachers experienced difficulties in identifying possible reasons for the behaviour manifested

by girls and boys. However, in the statements of most teachers, two main areas of causes can be noticed.

*Biological / psychological factors.* Among biological factors, teachers look for causes in the different brain functions of girls and boys, the different course of their adolescence, temperamental differences and differences related to motor activities.

> I recently read a great book called *Gender of the Brain* and there is some information about it. Even though we will bring up a boy and a girl in the same way, the differences will be somewhere due to the structure of the brain (T5).

> A woman thinks in a complex way. Why? What for? And the guy already knows: that's how it should be and that's it. (T7).

*Social factors.* Among the social factors mentioned by scientists that may affect the behaviour of male and female students, the influence of electronic media and mass media, the influence of early school education, traditional upbringing in the family and, more broadly, stereotypes rooted in culture are mentioned. It is worth pointing to the fact that only a small group showed awareness of the impact of their own stereotypical beliefs on the actions taken (which corresponded to the low popularity of the belief that the reasons for differences observed in the behaviour of boys and girls are, among others, ingrained and culturally transferred gender stereotypes). However, a few people pointed out this important aspect, self-awareness, and that no one is free from stereotypes and that in order not to treat schoolgirls and students stereotypically, constant self-reflection is necessary:

> Well, today I said to one boy during the lesson: 'My God, how are you scribbling! Look, you write like a doctor with twenty-five years of experience'. And now I think I said stereotypically that a doctor's scribble and prescription cannot be read. So it slipped out, I realised it today (T18).

**III. Strategies undertaken in response to boys' and girls' stereotypical behaviour.** Only one person did not take any action to deal with the differences in the behaviour of boys and girls. The rest made various efforts and had strategies in response to the perceived differences in youths' behaviour.

*Conversations about stereotypes.* Teachers mentioned strategies consisting of talking about stereotypes, encouraging boys and girls to go beyond schematic thinking and acting, and taking actions to eliminate the influence of stereotypes by universalising the experience of boys and girls as humanistic (with reference to social diversity). One of the most popular ways of dealing with differences in the behaviour of female and male students was talking about stereotypes, which appeared more often in humanities, in relation to the content of the curriculum:

> When we discuss 'Ballads and Romances' for the first time and talk about the ideal of a woman, described as an angel, gentle and fragile, the girls are outraged: 'How come?' And this is also a field of discussion about the fact that women have had rights for a little over a hundred years. Then I say: 'Girls, see what your great-grandmothers had to do' (T8).

The examples given by teachers, however, focused on social changes and the achievements of women's emancipation, as if ignoring the fact that stereotypes exist every day and we can observe them in the classroom. The lack of this connection made young people recognise the phenomenon of stereotyping as a concept that does not concern them directly because it relates to how it used to be and how it is in today's adult world. In reference to the equality of

women's and men's rights, which appeared in the context of discussing the causes of girls' and boys' different behaviour, the dominant message was that equal treatment means being treated in the same way (a reference to the outdated concept of equality rather than equity). The lack of reaction on the part of teachers to such an intuitive understanding of the concept of gender equality was noticeable.

*Encourage going beyond the box*. Another strategy was to encourage non-stereotypical behaviour and going beyond the well-established patterns of action within the behaviour stereotypically assigned to a specific gender. This happened both in response to specific behaviours in school or class and during conversations:

> I try to persuade girls all the time to be more go-getting because they really do sometimes have a task well calculated but do not come forward with a well-solved task at all (T3).

In the examples and ways of coping that the teachers spoke about in the interviews, the key was the lack of reference to the functioning gender stereotypes as a possible cause of the observed behaviours. The teachers did not refer to this fact, with consequences for their actions, which focused on individual coping strategies for, for example, shyness in girls or careless handwriting in boys. Sometimes the chosen strategies, although well intentioned, seemed controversial to say the least, as they could potentially backfire or lead to even greater antagonism between boys and girls:

> I ask the boys not to say anything at all. For example, when you need to run a maths app on their smartphones (. . .) I say, 'if anyone says anything, he'll get a negative assessment right away', and then I encourage the girls by asking which one will do it first (T3).

This is actually part of a strategy for apparent elimination of stereotypes, which focused on showing the same opportunities for boys and girls and not on differentiating behaviours or the requirements for them. The activities of teachers focused on showing social diversity, including that resulting from gender, but they did not take into account the existing stereotypes and the costs of the social non-conformity related to them as incurred by people who did not fit into gender stereotypes. Among the strategies, it was relatively popular to provide good examples (so-called role models) to show that it is possible to break gender stereotypes. This strategy was used both in the context of recalling historical figures to show a non-male perspective and overcome male domination in science and by referring to the teachers' own examples:

> First of all, I always tell them (the girls) that I am an engineer, so I also finished my technical studies; I made it. They can do it too. I tell them not to be afraid of it (T3).

This strategy can be effective, assuming that the selected examples are not simply tokenism [76]—i.e. exposing a person from a minority or disadvantaged group, which is supposed to prove equal status and in fact does not translate into a real change. The mere indication that there are women who do not fit the stereotypical image is definitely not enough to break the stereotypical perception of gender.

Summing up, it is worth appreciating the teachers' commitment and good intentions aimed at equal (in the sense of fair) treatment of boys and girls and the attempts to counteract the effects of social stereotyping of gender described by some teachers. However, it is hard not to notice that the described ways of dealing with situations where teachers identified a problem did not take place in a way in which the situation was openly named (referring to gender

stereotypes), which would have allowed the vicious circle of stereotypes and the self-fulfilling prophecy mentioned in the theoretical framework to be broken.

*Teachers' actions strengthening gender stereotypes.* It happened that the surveyed teachers were so attached to the 'delicate' image of girls that they hardly accepted their behaviour that deviated from the customary belief:

> I don't remember any more whether any girl said she watched a nice romance or something, even with the older ones. Only *The Witcher*, *Game of Thrones* or *James Bond* is what everyone watches. (T25)

This also occurred in terms of clothing, which, as they observed with dissatisfaction, tended towards unification. The image was even associated with difficulty in accepting a pattern of behaviour that differed from the current one with respect to one's own children. Forcing students into stereotypical roles also appeared in statements about them—for example, through quasi humorous phrases addressed to boys: 'a bunch of rascals, as I say to them' (T7). During the lessons, teachers declared that they tried to place students in groups based on their skills, but more often they declared that students decided how to arrange the teams. The choice of students was most often based on being the same gender.

Leaving students free to choose a group for work can strengthen gender polarisation, limiting the ability to develop skills in both girls and boys. Sometimes this situation is even exacerbated by giving children different tasks for each gender. When working in groups, for example, teachers offered girls reading topics that described only female characters while boys were given male characters. It is significant that several statements referred to the household in its traditional sense. Speaking of schoolgirls, teachers referred to roles that over the years have only been assigned to women: 'These girls are not too girly, I would say. None of them knit, none of them sew any clothes, but that's just the way it is' (T25). Convinced that they work for the benefit of students, they often referred to stereotypical gender roles fulfilled by adults:

> Girls give up a lot. For example, I say to a girl: try, you will also have to cope at home. If you don't find what you have in the recipe, maybe you will use a replacement sometimes, right? I encourage them to be open (. . .) (T23).

The respondents' statements also mentioned the stereotypical perception of girls as weaker. According to the teachers, they should be organised and cautious: 'For example, my youngest daughter, I also have to keep reminding her about everything because she, for example, will forget her sports outfit. And someone might say, "What a girl she is, always forgetting something"' (T10). The only feature that distinguished girls was their beauty: 'I always tell her (the female student), "You are so beautiful"' (T23).

**IV. Awareness of gender bias in textbooks.** *Lack of awareness of the existence of stereotypical content.* Most of the surveyed teachers did not pay attention to the way in which femininity and masculinity are presented in textbooks and also concluded that they had never considered such content deeply. In their statements, however, one can notice internalised and partially unconscious gender stereotypes:

> I don't know. I've never thought about it. I don't think I've ever noticed anything like that. As for the content of the task, I don't know—strong boys and here weak girls. Honestly, no, I didn't notice (T22).

Most often, teachers declared that the division of roles was addressed in the educational content, defining it as a 'partnership', and that the textbooks did not show attachment to the traditional image of roles considered female and male. They seem to have completely missed the way in which the subject was presented. In the interviews, there were also statements that testified to the deepening of stereotypes by teachers who assumed in advance which school reading would be more appropriate for boys and which for girls.

*Underestimating gender stereotypes*. It often happened that teachers downplayed the stereotypical approach to male and female roles in school literature. Sometimes they justified the gender roles attributed with reference to cultural and historical heritage. Thus, they deepened the 'traditional' view of the behaviours and activities attributed to girls and boys. It also happened that teachers justified the stereotypical approach to gender in textbooks by the need to convey the content properly: 'Because it's so hard to say that Pawel was buying a dress' (T7). Some of them downplayed and even deepened the mentioned approach by treating it carelessly and mockingly:

> In one task, we laughed, because of course it was about losing weight. And a woman was losing weight, and that poor woman couldn't lose weight. But in fact, there is no task about men losing weight (T19).

Despite the fact that the teachers often saw gender polarisation in the texts, they did not attach any importance to it, focusing only on the mathematical content. The Polish language teachers, on the other hand, if they saw differences in the presentation of gender roles in students' reading books, had a stereotypical approach to students' reading preferences.

*Noticing gender stereotypes*. Very rarely there were statements that proved teachers' awareness of the existence of stereotypical content. Moreover, gender polarisation in school reading books was definitely noticed more often by the Polish language teachers. The vast majority of teachers' statements concerned the presentation of girls and women and even their lack of their performance in school reading books. The respondents noticed that boys or men were almost always the heroes in books. It often happened that only male characters were discussed in lessons. Teachers sometimes tried to break this trend by also focusing students' attention on these rarely appearing female historical heroes:

> Emilia Plater, who died in defence of her homeland, and there are a few such heroines who really should be talked about (T16).

The surveyed teachers also pointed out that if female characters appeared in books, they were usually of little significance or required male help. The conversation during the interview also prompted the respondents to think more deeply about some issues regarding the approach to male and female roles. There were times when the teacher made a decision to pay more attention to these matters.

*Seeing non-traditional representations of gender*. Few—in fact, only two statements—showed that the non-stereotypical perception of gender roles was noticed in textbooks:

> There is an interesting text in fourth grade: Dad cooks, cleans and looks a bit like mommy because he has a braid and so on. But he cooks best when mum goes to the pool. Dad cooks and looks after the baby (. . .) So somewhere here is the beginning of this detachment from the stereotype (T12).

More often, however, teachers noticed non-stereotypical content but they were not fully convinced about its importance and significance. They seem to have educated themselves through the writing in textbooks.

## Discussion and conclusions

This study was designed to investigate: (i) teachers' awareness of gender-stereotyped behaviours of girls and boys; (ii) teachers' awareness of the possible causes of these behaviours; (iii) teachers' actions to respond to these behaviours, including those that may deepen gender bias and gender stereotypes- and (iv) teachers' awareness of the existence of gender-polarised content in school textbooks.

As it turns out, educators, with minor exceptions, are aware of the differences in the functioning of girls and boys at school. Unfortunately, it appears that they seem to have no awareness about the stereotype threat phenomenon [48, 49]. Teachers directly indicated the gender division of fields, assigning better mathematical skills to boys and language skills to girls. Moreover, they seemed to accept this state of affairs without reflection, despite belonging to a marginalised gender in the case of the field of mathematics. Meanwhile, as Sanders [77] has rightly emphasised, the task of teachers should be to recognise and extinguish gender bias in the classroom, as it may particularly limit the ambitions and achievements of female students. Among other stereotypical gender behaviours, teachers also mention higher emotionality and willingness to cooperate as belonging to a group of behaviours manifested by girls, as well as their tendency to strive for perfection with a simultaneous lack of self-confidence. In turn, among boys, the need for agency, rationality and the will to compete and dominate were identified. A greater behaviour problem was also emphasised among boys, and a higher level of care and good behaviour was noted among girls.

The actual reasons for these manifested behaviours, however, remain beyond teachers' awareness and knowledge. Teachers have great difficulty identifying the causes of the behaviours that occur, or they look for them in biological factors that characterise the students themselves, or in the influences of the family and social environment. Only a small group of teachers actually notice the influence of their own beliefs on the stereotypical functioning of students.

Teachers undertake a number of behaviours in response to observed student behaviour. Sometimes there are discussions about gender stereotypes on the basis of the discussed source texts; teachers also encourage students (mainly girls) to go beyond the usual patterns of functioning. Unfortunately, as shown by the results of the interview part of the study, possible attempts undertaken by teachers to eliminate gender stereotypical behaviours may sometimes lead to the opposite situation in which gender polarisation only deepens. The observations of the lessons also show that during classes, the traits stereotypically attributed to girls (ability to cooperate) and boys (willingness to compete) are systematically strengthened and rewarded by teachers, and boys, who are more often reprimanded for disturbing behaviour, seem to be additionally favoured by frequency of contact with the teacher. Meanwhile, for teachers' interventions to be effective, the subject of stereotypes should be covered. They should break the vicious circle of stereotypes [15] or show a different strategy to break stereotypical behaviour among students.

Finally, awareness of gender-polarised content in school textbooks is negligible among teachers. Teachers mostly do not pay attention to the content of tasks and texts, emphasising only their substantive value. If gender-polarised content is noticed, it is usually ignored or considered to be of little importance to the learning process. A small number of educators, mainly teachers of the Polish language, notice the stereotypical representation of women and men in

the teaching content, sometimes even emphasising the content that shows gender in a non-stereotypical way. However, this seems to be the exception that proves the rule.

The situation in education in Poland has been constantly reformulated since the beginning of the 1990s. In 1999, a comprehensive, structural reform of education was undertaken; in 2007, another reform covered the curriculum and the evaluation system; and 2014 brought a reform of early childhood education. With the 2017 reform, a major structural change was again brought to the pre-1999 state of affairs [78]. The recent ministerial changes initiated with the introduction of reform in 2017 are revolutionary rather than evolutionary. With the introduction of the 2017 reform, the curricula were changed again; the year 2021 brought additional curricula; and now, in 2022, new changes to the structure of the system have been proposed, giving the Boards of Education a decisive role [79]. Despite introducing amendments to the curricula and the reforms implemented in a relatively short period of time, virtually none of the changes took into account broadly understood gender issues [80].

This does not mean, however, that these issues do not have a significant impact on the subject of our research; the situation is quite the reverse. Teachers' lack of knowledge about the risk of exacerbating gender stereotypes, resulting from their insufficient education, actually strengthens the stereotypes [81]. This situation is also aggravated by the highly polarised teaching content on which teachers rely for their work with students [37]. The above factors seem to aggravate the already existing inequalities between male and female students instead of reducing them and showing students the strength and benefits of equality. It has been known for a long time that gender stereotypes are one of the causes of social inequality. In the process of preparation for the teaching profession in Poland, however, there is no reference to this issue at any time [11, 82]. It is difficult to expect teachers to efficiently counteract the limiting influence of gender stereotypes since they do not have an opportunity to learn how to recognise and react to these stereotypes. Knowing and understanding how this situation is perceived by teachers, in addition to the obvious scientific value, can help in designing adequate educational interventions for them, the aim of which is to teach educators how to counteract the limitations resulting from the influence of gender stereotypes. In fact, as research practice shows, such interventions prepared for teachers are highly effective and can significantly improve the situation for girls and boys participating in the teaching process. An example of such an intervention is the teacher training programme REFLEC, aimed at strengthening teachers' competences to support female and male students to develop their potential without the constraints of gender stereotypes [83]. Of course, as our study was mainly qualitative, its results cannot be generalised to the entire group of Polish teachers. The authors are also aware that the same-sex sample of teachers is quite homogeneous, and it is impossible to infer on this basis about the entire population of teachers of the above subjects, or to make a gender comparison. However, the results of the present study echo a certain trend of repetitive patterns of teacher behaviour present in the results of other studies in this area, which we mention in the theoretical framework. We are also aware of a lack of real control over the curriculum for prospective teachers, which we only mention in the context of the lack of content pertaining to gender issues. Teachers were not asked directly about their knowledge of gender differences and gender issues obtained during their education process; this knowledge is taken only from the literature. However, we assume, taking into account the age of the surveyed teachers and the most recent data from the reports on teacher training in this area [11, 82], that little has changed over the last ten years.

Considering the above, in future research, it would be worth ensuring a comparable number of male and female teachers teaching both mathematics and language (humanities) subjects. Moreover, it would be worth comparing the achieved study results with the analysis of contemporary curricula in teacher training. Nevertheless, our research shows the issue of

breaking gender stereotypes should be given much more attention because, as shown by the situations observed in classrooms, they often lead to the strengthening of gender stereotypes. When designing educational interventions, and even earlier in the preparation of curricula for prospective teachers, it is very important to consider this issue.

## Author Contributions

**Conceptualization:** Aleksandra Gajda.

**Data curation:** Aleksandra Gajda.

**Formal analysis:** Aleksandra Gajda, Agnieszka Bójko, Ewa Stoecker.

**Funding acquisition:** Aleksandra Gajda.

**Investigation:** Aleksandra Gajda.

**Methodology:** Aleksandra Gajda.

**Project administration:** Aleksandra Gajda.

**Writing – original draft:** Aleksandra Gajda, Agnieszka Bójko, Ewa Stoecker.

**Writing – review & editing:** Aleksandra Gajda, Agnieszka Bójko, Ewa Stoecker.

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
