## [Decision Letter · Decision Letter 0]

15 Feb 2022

PONE-D-21-37263The Vicious Circle of Stereotypes: Teachers’ Awareness of and Responses to Students’ Gender-Stereotypical BehaviourPLOS ONE

Dear Dr. Gajda,

Thank you for submitting your manuscript to PLOS ONE. After careful consideration, we feel that it has merit but does not fully meet PLOS ONE’s publication criteria as it currently stands. Therefore, we invite you to submit a revised version of the manuscript that addresses the points raised during the review process.

We look forward to receiving your revised manuscript.

Kind regards,

Hubert János Kiss

Academic Editor

PLOS ONE

Journal Requirements:

“The preparation of this article was supported by grant UMO-2018/29/B/HS6/00036 to Aleksandra Gajda from the National Science Centre Poland.”

 “AG

grant no. 2018/29/B/HS6/00036

Polish National Science Centre

https://www.ncn.gov.pl/

Additional Editor Comments (if provided):

As you can see in the reports, one of the reviewers recommends major revision, while the other rejects the paper. However, the latter reviewer is also sympathetic toward the paper, but thinks that there is a lot to be done. I made the decision to give you the opportunity to revise the paper. I ask you to consider the concerns that were raised very carefully and take the reviewers' suggestions seriously. Reviewer 2 provided very specific advices on how to improve the paper. Please, consider them, and I believe that the paper will improve.

Reviewers' comments:

Reviewer's Responses to Questions

**Comments to the Author**

1. Is the manuscript technically sound, and do the data support the conclusions?

Reviewer #1: No

Reviewer #2: Yes

2. Has the statistical analysis been performed appropriately and rigorously? 

Reviewer #1: No

Reviewer #2: Yes

3. Have the authors made all data underlying the findings in their manuscript fully available?

Reviewer #1: No

Reviewer #2: No

4. Is the manuscript presented in an intelligible fashion and written in standard English?

Reviewer #1: Yes

Reviewer #2: Yes

5. Review Comments to the Author

Reviewer #1: I have carefully read the manuscript I have received to review (Manuscript Number: PONE-D-21-37263 entitled “The Vicious Circle of Stereotypes. Teachers’ Awareness of and Responses to Students’ Gender-Stereotypical Behaviour”).

The work is an interesting effort to explore the relationships between teachers` gender related beliefs and attitudes and their actual behavior in classroom. The authors’ use of a unique data set with combined interview and observation data could be a major strength of the manuscript.

The topic is psychologically, socially and even politically important and necessary. But regardless of that, I have some concerns with the manuscript in its present form, especially regarding the fit between the theoretical framework and the methods used. Moreover, I miss clear research questions.

My main concern is that the introduction section does not lead to clear reseach questions and that the theoretical aspects covered in the introduction section seem disconnected from the methods of the study.

For example, the interview questions, derived themes and also the categories for the obserevations seem unrelated to the contents of the introduction (especially the gender bias in textbooks never turns up before the present study section). It would be important to write an introduction that logically leads to questions to be asked and behaviour aspects to observe. For example the authors should adress the content of gender stereoytpes (agency, communion) in the introduction section and not only in the results section. I think this is especially important when a grounded theory approach is taken to analyse the interviews - the theory the analysis is grounded on should be the same as the one in the introduction section of the manuscript. Also, there are parts of the introduction that seem to have no relevance at all for the present study, e.g. the paragraph covering biological gender differences.

My second major concern is the lack of clear research questions and a clear vision of how the two different data types can be tied together.

In addition, the statistical analysis of the observational categories is not described in enough detail. There is no information about the total number of codings, about the coding units, about interreater reliability and so on. Moreover, the authors seem to have conducted a series of t-tests without controlling for the familywise error, thus neglecting the problem of alpha cumulation.

Minor comments:

I would suggest having the "participants" and "procedure" sections seperate.

Why were there less teachers than classes? Were some teachers teaching more than one class?

There should be a clear description of how the interview questions were posed. At the moment the formulation of the questions sounds very academic. I wonder, if it's clear to teachers what "awareness of the existence of different, stereotypical behaviours presented by male and female students" means.

The categories for the obersvational findings should be part of the method section.

Results and Discussion should be seperate sections

In the results section I miss information about the duration of the interviews, the coding process, coding units, number of codes, interrater reliability etc.

I won't go into detail regarding the whole results section but I see several problematic interpretations there and some answers that I do not understand, e.g.

"Well, I haven't had such a brilliant Cinderella in my life. The mere fact that he had

already changed, mum dressed him there, because he was such a modern Cinderella,

mum dressed him in a dress, he simply played sensational, wonderful, he was

applauded very much"  I don't even know what that means.

"Female and Male School Subjects" should be "Stereotypically feminine and masculine school subjects"

In the Conclusion section a new theoretical perspective (stereotype threat) turns up that wasn't mentioned before.

Reviewer #2: The paper uses interview data so it is understandable that the data is not made publicly available. All my comments related to the content of the paper are in the attached file. I wish the authors good luck with their manuscript.

6. PLOS authors have the option to publish the peer review history of their article (what does this mean?). If published, this will include your full peer review and any attached files.

Reviewer #1: No

Reviewer #2: No

---

## [Author Response · Author response to Decision Letter 0]

4 Mar 2022

First Reviewer's Comments

Comment 1: The work is an interesting effort to explore the relationships between teachers` gender related beliefs and attitudes and their actual behaviour in classroom. The authors’ use of a unique data set with combined interview and observation data could be a major strength of the manuscript. 

Response 1: Thank you very much for appreciating the idea of our research scheme.

Comment 2: The topic is psychologically, socially and even politically important and necessary. But regardless of that, I have some concerns with the manuscript in its present form, especially regarding the fit between the theoretical framework and the methods used. Moreover, I miss clear research questions

Response 2: At the end of the 'introduction' section, four research objectives were formulated, which correspond to the research questions formulated in the 'Method' section. The theoretical framework has been thoroughly reformulated so that the narrative clearly shows the rationale for the research questions posed.

Comment 3: My main concern is that the introduction section does not lead to clear research questions and that the theoretical aspects covered in the introduction section seem disconnected from the methods of the study.

For example, the interview questions, derived themes and also the categories for the observations seem unrelated to the contents of the introduction (especially the gender bias in textbooks never turns up before the present study section). It would be important to write an introduction that logically leads to questions to be asked and behaviour aspects to observe. For example, the authors should address the content of gender stereotypes (agency, communion) in the introduction section and not only in the results section. I think this is especially important when a grounded theory approach is taken to analyse the interviews - the theory the analysis is grounded on should be the same as the one in the introduction section of the manuscript. Also, there are parts of the introduction that seem to have no relevance at all for the present study, e.g. the paragraph covering biological gender differences 

Response 3: Both the first and the second reviewer pointed to the need to reformulate the theoretical framework of the article. In the current version, an attempt was made to structure the content in such a way that it logically leads to the deduced goals of the research (cited at the end of the theoretical framework) and research questions. A paragraph describing gender biased textbook content has also been added and the description of gender stereotypes has been extended.

The comment on the lack of relationship between biological differences and the subject of the article is also consistent with the comments of the second reviewer. We decided to remove this paragraph from the article, wishing to focus mainly on the socio-cultural determinants.

Comment 4: My second major concern is the lack of clear research questions and a clear vision of how the two different data types can be tied together 

Response 4: The objectives of the study were clarified, as well as the research questions, currently included in the method section. It was also explained which research questions correspond to the observational part and which to the interview part. We hope that this wording of the goals is clear and leaves no doubt about the combination of two data types in one text. We want to emphasize our gratitude to this element of the review, we agree that the article has crept into chaos in formulating research goals and questions.

Comment 5: In addition, the statistical analysis of the observational categories is not described in enough detail. There is no information about the total number of codings, about the coding units, about interreater reliability and so on. Moreover, the authors seem to have conducted a series of t-tests without controlling for the familywise error, thus neglecting the problem of alpha cumulation 

Response 5: The description of statistical analyzes concerning the observations was supplemented. In the current text, two tables have been added, informing about code categories, examples of behavior and interrater reliability (Table 2), as well as the number of coded behaviors in each category (Table 4). The information on familywise error has also been supplemented.

Comment 6: I would suggest having the "participants" and "procedure" sections separate 

Response 6: Thank you for this comment, the sections have been separated as suggested.

Comment 7: Why were there less teachers than classes? Were some teachers teaching more than one class? 

Response 7: Some teachers taught in more than one class, hence the difference in the number of observed classes and the number of teachers surveyed. A related explanation has been added in the text in the ‘Participants’ section

Comment 8: There should be a clear description of how the interview questions were posed. At the moment the formulation of the questions sounds very academic. I wonder, if it's clear to teachers what "awareness of the existence of different, stereotypical behaviours presented by male and female students" means. 

Response 8: We agree with the reviewer that we missed out on detailed interview questions. This has been completed. In addition, Table 1 was added to the article for clarity.

Comment 9: The categories for the observational findings should be part of the method section. Results and Discussion should be separatee sections

Response 9: The categories for the observational findings were moved, as suggested, to the 'method' section. The Results and the Discussion were also separated. Thank you for highlighting these points.

Comment 10: In the results section I miss information about the duration of the interviews, the coding process, coding units, number of codes, interrater reliability etc. 

Response 10: Information on the duration of the interviews was previously included in the 'study procedure' section. In the 'data analysis' section, we added, as suggested by the reviewer, information on coding process, coding units, number of codes, interrater reliability etc. Thank you very much for pointing out the lack of information that needed to be supplemented.

Comment 11: I won't go into detail regarding the whole results section but I see several problematic interpretations there and some answers that I do not understand, e.g.

"Well, I haven't had such a brilliant Cinderella in my life. The mere fact that he had

already changed, mum dressed him there, because he was such a modern Cinderella,

mum dressed him in a dress, he simply played sensational, wonderful, he was

applauded very much"  I don't even know what that means 

Response 11: The illegible quote that the reviewer underlines has been removed. In addition, the entire 'Results' section was reformulated in an attempt to shorten it and reduce the number of citations, while maintaining the sense of the statements and support for the summaries.

Comment 12: "Female and Male School Subjects" should be "Stereotypically feminine and masculine school subjects" 

Response 12: Thank you for this suggestion, a corresponding revision has been made.

Comment 13: In the Conclusion section a new theoretical perspective (stereotype threat) turns up that wasn't mentioned before Response 13: Both reviewers emphasize this point, thank you for drawing attention to it. The subject of the stereotype threat has now been introduced in the theoretical part of the article.

Second Reviewer's Comments

Comment 1: Thank you for the opportunity to review this paper. The data that are gathered are impressive and the mixed method combination fits the goals of the paper. I think the topic is very relevant in light of boys’ underperformance in education, gender segregation in education and other problems that arise from the reinforcement of gender stereotypes in schools. Although there were parts of the paper that I liked, I think there is also room for improvement. My comments can be found below, the most important ones are in bold. 

Response 1: We would like to thank the reviewer for the kind words and appreciation of the idea for our research scheme. We are very grateful for your comments, we hope that the text will improve after their introduction.

Comment 2: Please mention in abstract what type of education is studied (primary/secondary school). 

Response 2: Appropriate addition has been added in the abstract.

Comment 3: “Despite the passage of time, these social beliefs about gender roles have remained remarkably stable”→ do you have a more recent reference? 

Response 3: The text was supplemented with more up-to-date references.

Comment 4: I miss a) why we want to know this (some reasons are touched upon on page 13 but this should come sooner/be worked out more), b) what has previous research found regarding the questions you ask/goals you set c) linking to b: what is the innovation of this study? 

Response 4: We thank the reviewer for this comment. In response, the "Increasing gender stereotypes at school" section was supplemented with the results of previous research in relation to the research objectives. An innovative element has also been indicated, which may fill the knowledge gap in this field.

Comment 5: I miss references to key works the social construction of gender such as West & Zimmerman (1987). Also: take a look at the work of Mieke van Houtte. 

Response 5: We greatly appreciate this comment on the references. Information on the social construct of gender has been included in the new section 'School as an environment of socialization to gender roles'. I also used articles by Mieke van Houtte, which are quoted in the section 'Increasing gender stereotypes at school'

Comment 6: In the part on biological differences between girls/boys the conclusion is drawn that both biological/social factors should be taken into account. How is that done in this study? 

I would be careful with referring to Gurian and Sommers as evidence for the fact that brains differ between girls and boys; there are also studies that show there are very little differences or that brain differences actually emerge from social inequalities. At least mention this is not an undisputed fact. See e.g. these works: 

1. Wierenga, L. M., Bos, M. G., van Rossenberg, F., & Crone, E. A. (2019). Sex effects on development of brain structure and executive functions: greater variance than mean effects. Journal of cognitive neuroscience, 31(5), 730-753. 

Kleinherenbrink, A. (2014). Mapping plasticity: Sex/gender and the changing brain. Tijdschrift voor Genderstudies, 17(4), 305-326. Response 6: This comment also corresponds to the comment of the first reviewer, thank you for drawing attention to this issue. After rereading this section, we decided to remove the information regarding biological differences as research in this topic is contradictory. Our research deals with social and cultural determinants and this is what we ultimately focused on in the theoretical framework.

Comment 7: In sum: The part before “the present study” is a quite unstructured literature review. It has some nice parts, but I really miss a logical and concise build-up to certain expectations regarding the goals over the paper. E.g. there is quite some elaboration about concepts that are not directly and/or clearly related to the goals of the study. 

Response 7: The authors made every effort to respond to comments regarding the theoretical framework of the article. We hope that the current, thoroughly reformulated version will meet the expectations of the reviewers.

Comment 8: I would like more explanation of why the specific these forms of data gathering were used. What is the advantage of doing observations/interviews (in light of your goals)? What questions were answered by observations and what questions by interviews? I sometimes miss the clear link between introduction (goals), theory (see previously: what are the expectations based on previous studies and theory), method (what method is used for what question/goal/expectation). 

Response 8: The comment regarding the lack of connection of the theoretical framework with the method and the results presented in the text corresponds with the comment of the first reviewer. Thank you very much for drawing attention to this shortcoming. The objectives of the research were clearly specified in the 'Introduction' section and research questions in the 'Method' section. It was also explained which research questions correspond to the observation part and which to the interview part. An explanation has also been added regarding the choice of these research methods in the ‘Present study’ section.

Comment 9: - Some reflection on how teacher’s behaviour might be altered because of a) the presence of the researcher and b) knowledge about what the study was about (it is not mentioned what information was given to the teacher before observation). 

- I am not a qualitative researcher but the (method of) analysis seems thorough and sound. 

Response 9: The "Study procedure" section has been supplemented with information explaining how the impact of the presence of researchers on the functioning of students and teachers during observation was reduced. It also describes what information about the study was given to teachers during the observation phase.

Comment 10: As said, I am not a qualitative researcher but I think the result section is far too long and should be more focussed. In total it’s 18 pages (there are full articles that are shorter) and there are 7 pages alone on describing behaviour patterns that differ between girls and boys (observed by teachers). I appreciate the authors wanting to present all these interesting quotes, but it makes the paper not well readable and makes the reader lose track of what the goals are and how these qualitative results add to these goals. 

Ways to shorten and structure the result section: There are often multiple quotes to illustrate 1 point and some quotes are very long (e.g. 3 long quotes on page 27). There are also quotes that do not add to the text at all (eg. the quote on the intimate relation with the teacher, what is the goal/context of this quote?). General conclusions (sh/)could be drawn at the end of a section so that the link to the goal is clear. My suggestion would be to seriously reduce the number of quotes, focus the text so that it clearly relates to the goals of the study (e.g. draw clear conclusions about the different goals at the end of the result sections). 

Response 10: As suggested by the reviewer, the part describing the results of the study was thoroughly shortened. The number of citations has been reduced, and citations not related to the topic of research questions have also been removed. However, the short summary of each part was abandoned in favour of a broader summary, which was included in the discussion. We hope that in the current version the 'results section' is more readable and corresponds to the structure of the research areas.

Comment 11: Most importantly: I miss a structured reflection on the 4 goals of the paper. 

Response 11: The discussion was thoroughly reformulated, emphasizing reflection on the four main objectives of the study and the results obtained in the two phases of the study.

Comment 12: Could an explanation for boys receiving more attention from the teacher be that they are underperforming (and teachers spend more time on underperforming students)? 

Response 12: Yes, it is one of the most common explanations for this, and it also appears frequently in the literature (Sadker, 2000; Younger et al., 1999). Appropriate reference has been added.

Comment 13: Small: this sentence is unclear: “Meanwhile, in order for teachers' interventions to be effective, they should relate, and more specifically, break the vicious circle of stereotypes (Pankowska, 2005) or show a different strategy of breaking stereotypical behaviour among students.”→ for teachers interventions to be effective (to break gender stereotypes?) they should break the vicious circle of stereotypes? 

Response 13: The sentence that was unclear was paraphrased as follows:

‘Meanwhile, for teachers' interventions to be effective, the subject of stereotypes should be covered. They should break the vicious circle of stereotypes (Pankowska, 2005) or show a different strategy of breaking stereotypical behavior among students.’

Comment 14: I would not spend a substantial part of the conclusion introducing a theoretical concept (“activation of a negative stereotype about the low ability level of an individual’s own group”). 

Response 14: A similar comment was proposed by the first reviewer. We decided to shorten the description of this phenomenon in the discussion and pay more attention to it in the theoretical part of the article.

Comment 15: There is mentioning on changes in education in Poland but these are not explained. What is meant by this?: “The situation of education in Poland since the beginning of the nineties of the last century has constantly been reformulated (Jakubowski, 2021), and the recent ministerial changes initiated with the introduction of another education reform in 2017 are revolutionary rather than evolutionary (Ministry of Education and Science, 2021).” 

Response 15: A supplement was added to the discussion, briefly describing successive changes in the education system in Poland over the last three decades. The authors trust that in the current version it will be more understandable for the reader.

Comment 16: There could be more suggestions for future research 

Response 16: As suggested by the reviewer, the excerpt describing the suggestions for future research has been extended

Comment 17: Would the fact that you only surveyed female teachers have influenced the findings? There could be some reflection on this. 

Response 17: This is a very valid comment, relevant comment was added in the limitations paragraph. Thank you very much for highlighting the issue that we missed.

---

## [Decision Letter · Decision Letter 1]

23 Mar 2022

PONE-D-21-37263R1The Vicious Circle of Stereotypes:

Teachers’ Awareness of and Responses to Students’ Gender-Stereotypical BehaviourPLOS ONE

Dear Dr. Gajda,

Thank you for submitting your manuscript to PLOS ONE. After careful consideration, we feel that it has merit but does not fully meet PLOS ONE’s publication criteria as it currently stands. Therefore, we invite you to submit a revised version of the manuscript that addresses the points raised during the review process.

We look forward to receiving your revised manuscript.

Kind regards,

Hubert János Kiss

Academic Editor

PLOS ONE

Journal Requirements:

Additional Editor Comments:

Dear Authors,

Reviewer 1 sent her comments regarding the revision that you prepared. Reviewer 2 was too busy to check whether her comments have been taken into account, so I read the paper myself.

Overall, my judgment about the new version coincides with Reviewer 1's opinion. You considered seriously the points raised by the reviewers, and, as a result, the study improved. However, still there is room for some improvement, as Reviewer 1 states clearly in her second review. Fortunately, the changes that she proposes are minor, so my decision this time is minor revision. Please, consider the suggestions of Reviewer 1 and submit a new version.

Reviewers' comments:

Reviewer's Responses to Questions

**Comments to the Author**

1. If the authors have adequately addressed your comments raised in a previous round of review and you feel that this manuscript is now acceptable for publication, you may indicate that here to bypass the “Comments to the Author” section, enter your conflict of interest statement in the “Confidential to Editor” section, and submit your "Accept" recommendation.

Reviewer #1: (No Response)

2. Is the manuscript technically sound, and do the data support the conclusions?

Reviewer #1: Yes

3. Has the statistical analysis been performed appropriately and rigorously? 

Reviewer #1: Yes

4. Have the authors made all data underlying the findings in their manuscript fully available?

Reviewer #1: No

5. Is the manuscript presented in an intelligible fashion and written in standard English?

Reviewer #1: Yes

6. Review Comments to the Author

Reviewer #1: I thank the authors for their thorough revision of the manuscript and their answers to my comments.

In my opinion, the manuscript has significantly improved. However, I still have some remarks:

1. In the abstract, I would change "a self-fulfilling prophecy" to "self-fulfilling prophecies".

2. In the section about sex and gender the authors define sex as "anatomical and psychological features" - I don't agree with that. Also I wonder about the reference here. Why don't you cite Kay Deaux or West & Zimmerman?

3. I would add agency/communion (or instrumentality/expressivity) as the theoretical dimensions behind gender stereotypical attributes, see e.g. Kachel et al. (2016) or Abele & Wojciszke (2007) or

4 I suggest to change the order of the research questions so that it matches the logic of the paper. So first pose thequestion that is answered through observation and then add the questions that are answered through the interviews. Moreover, there shouldn't be a RQ 3.1. if there is no 3.2.

5. Please add sample statements to all categories presented in the Results & Discussion section (e.g. Biological / psychological factors)

6. References to the literature from the introduction section are missing from the Results & Discussion Section. The authors should decide if they want a pure Results section (without references) or a Results and Discussion section than really discusses the results with regard to the literature.

7. In the Conclusion section, I suggest to change the wording here "The presented study was designed to verify; (i) teachers' awareness of gender stereotyped behaviours of girls and boys, (ii) teachers' awareness of the possible causes of these behaviours, (iii) teachers' actions to respond to these behaviours, including those that may deepen gender bias and gender stereotypes, and (iv) teachers' awareness of the existence of gender-polarized content of school textbooks." In my opinion, it's not about verifying but about investigating.

8. Please mention existing interventions that could improve the situation, e.g. Kollmayer et al. (2020)

Moreover, I proof-reading the whole manuscript for language and spelling errors and sentences that got mixed up during the revision (especially in the Results & Discussion section).

7. PLOS authors have the option to publish the peer review history of their article (what does this mean?). If published, this will include your full peer review and any attached files.

Reviewer #1: No

---

## [Author Response · Author response to Decision Letter 1]

5 May 2022

First Reviewer's Comments

Comment 1: I thank the authors for their thorough revision of the manuscript and their answers to my comments. In my opinion, the manuscript has significantly improved. However, I still have some remarks:

Response 1: We would like to thank the reviewer for this comment. This time also we will do our best to improve the article.

Comment 2: In the abstract, I would change "a self-fulfilling prophecy" to "self-fulfilling prophecies". Response 2: The change was introduced as suggested by the reviewer.

Comment 3: In the section about sex and gender the authors define sex as "anatomical and psychological features" - I don't agree with that. Also I wonder about the reference here. Why don't you cite Kay Deaux or West & Zimmerman? 

Response 3: The sentence “anatomical and psychological features” has been removed from the section. We would like to note that West & Zimmerman is quoted in this paragraph as suggested in a previous round of reviews (item 11: West C, Zimmerman, DH. Doing gender. Gender & society. 1987; 1 (2): 125-151) while one of Kay Deaux's books is also quoted (item 4 in reference list: Kite M, Deaux K, Haines EL. Gender stereotypes. In: Denmark FL, Paludi MA, editors. Psychology of women: A handbook of issues and theories. Santa Barbara, CA: Praeger an imprint of ABC-CLIO, LLC; 2017. p. 205-36.)

Comment 4: I would add agency/communion (or instrumentality/expressivity) as the theoretical dimensions behind gender stereotypical attributes, see e.g. Kachel et al. (2016) or Abele & Wojciszke (2007) Response 4: As suggested by the reviewer, the paragraph on stereotypical male and female features was supplemented with the proposed dimensions.

Comment 5: I suggest to change the order of the research questions so that it matches the logic of the paper. So first pose the question that is answered through observation and then add the questions that are answered through the interviews. Moreover, there shouldn't be a RQ 3.1. if there is no 3.2. 

Response 5: As suggested by the reviewer, the order of the research questions was changed to better correspond to the order in which the research results were presented.

Comment 6: Please add sample statements to all categories presented in the Results & Discussion section (e.g. Biological / psychological factors) 

Response 6: Where necessary, quotes from teachers' statements were added to the analytical categories. Thank you for highlighting this element. We missed this issue when shortening and reformulating the previous version of the text.

Comment 7: References to the literature from the introduction section are missing from the Results & Discussion Section. The authors should decide if they want a pure Results section (without references) or a Results and Discussion section than really discusses the results with regard to the literature. 

Response 7: Thank you for this comment, we decided to leave the Results section pure and move the discussion to the end of the article, with references to the literature.

Comment 8: In the Conclusion section, I suggest to change the wording here "The presented study was designed to verify; (i) teachers' awareness of gender stereotyped behaviours of girls and boys, (ii) teachers' awareness of the possible causes of these behaviours, (iii) teachers' actions to respond to these behaviours, including those that may deepen gender bias and gender stereotypes, and (iv) teachers' awareness of the existence of gender-polarized content of school textbooks." In my opinion, it's not about verifying but about investigating. 

Response 8: Thank you for highlighting this point. We agree, the proposed change has been made.

Comment 9: Please mention existing interventions that could improve the situation, e.g. Kollmayer et al. (2020) 

Response 9: Thank you for your suggestion to mention existing interventions. In line with this commentary, the necessary addition has been made to the text.

Comment 10: Moreover, I proof-reading the whole manuscript for language and spelling errors and sentences that got mixed up during the revision (especially in the Results & Discussion section). Response 10: The article was returned to proofreading. We hope that in its current version it meets the stylistic and grammatical requirements.

---

## [Editor Report · Decision Letter 2]

13 May 2022

The Vicious Circle of Stereotypes:

Teachers’ Awareness of and Responses to Students’ Gender-Stereotypical Behaviour

PONE-D-21-37263R2

Dear Dr. Gajda,

We’re pleased to inform you that your manuscript has been judged scientifically suitable for publication and will be formally accepted for publication once it meets all outstanding technical requirements.

Kind regards,

Hubert János Kiss

Academic Editor

PLOS ONE
---

## [Editor Report · Acceptance letter]

19 May 2022

PONE-D-21-37263R2 

The vicious vircle of stereotypes: Teachers’ awareness of and responses to students’ gender-stereotypical behaviour 

Dear Dr. Gajda:

I'm pleased to inform you that your manuscript has been deemed suitable for publication in PLOS ONE. Congratulations! Your manuscript is now with our production department. 

Kind regards, 

on behalf of

Dr. Hubert János Kiss 

Academic Editor

PLOS ONE